# Long-Term Results of Pediatric Liver Transplantation for Progressive Familial Intrahepatic Cholestasis

**DOI:** 10.3390/jcm11164684

**Published:** 2022-08-11

**Authors:** Chenyue Hang, Yijie Jin, Yi Luo, Mingxuan Feng, Tao Zhou, Jianjun Zhu, Jianjun Zhang, Yuan Liu, Qiang Xia

**Affiliations:** 1Department of Liver Surgery, Ren Ji Hospital, School of Medicine, Shanghai Jiao Tong University, Shanghai 200127, China; 2Shanghai Engineering Research Center of Transplantation and Immunology, Shanghai 200127, China; 3Shanghai Institute of Transplantation, Shanghai 200127, China

**Keywords:** PFIC, liver transplantation, steatosis, catch-up growth, intelligence development

## Abstract

We analyzed the long-term survival rate and development of progressive familial intrahepatic cholestasis (PFIC) patients after liver transplantation (LT). From October 2007 to May 2019, 41 patients were diagnosed as PFIC (type I-III) and received LT in Ren Ji Hospital due to end-stage liver diseases. The median age at LT was 2.93 years, with 75.6% of patients receiving living donor liver transplantation (LDLT). The 5- and 10-year patient survival rates after LT were 92.7% and 92.7%, respectively, and no difference was found among the three subtypes of PFIC. Two PFIC type II patients received re-transplantation due to vascular complications. Liver function and bile acid metabolism returned to normal levels in all living recipients. Catch-up growth was recorded as the height and weight Z scores increased from −2.53 and −1.54 to −0.55 and −0.27 with a median follow-up time of 5.55 years. Improved psychomotor ability and age-appropriate study ability was also observed. A total of 72.4% of school-aged recipients exhibited average academic performance. Diarrhea was reported in all PFIC type I recipients but resolved after resin absorptive treatment. However, allograft steatosis occurred in one PFIC type I patient and exhibited a “remission–relapse circle” under the treatment of cholestyramine. In conclusion, LT is an effective treatment for end-stage PFIC patients with encouraging long-term survival rate and development. However, allograft steatosis should be closely monitored in PFIC type I patients even if diarrhea has been well treated.

## 1. Introduction

Progressive familial intrahepatic cholestasis (PFIC) is a heterogeneous group of liver disorders of autosomal recessive inheritance, presenting as intrahepatic cholestasis in infancy or early childhood and resulting in end-stage liver disease (ESLD) [1]. Based on genetic mutations and clinical manifestations, PFIC is now divided into six subtypes, which are type I to VI [2,3]. All subtypes of PFIC are caused by defects in intracellular bile acid transport, secretion, and homeostasis of bile canalicular function [4,5]. The clinical presentations of PFIC vary from early-onset severe liver disease to episodic late-onset occurrence [6]. Among them, PFIC type I and type II patients generally present jaundice and severe pruritus in the first few months of life and may progress to ESLD in early childhood [7].

Treatments for PFIC patients include dietary supplement, drug treatment, and surgical interventions, such as biliary diversion [6]. Nutritional rehabilitation after meticulous nutritional assessment in PFIC patients includes sufficient calories and protein intake, as well as a dietary supplement of medium chain triglycerides and fat-soluble vitamins [6]. However, for patients who have progressed to ESLD or are suffering from unalleviated pruritus, liver transplantation (LT) is the only option. With the development of surgical skills and perioperative management, the survival rate and life quality of PFIC patients after LT has significantly improved. As reported, the 10-year patient survival rate after LT ranged from 72.7% to 90.9%, varying among PFIC subtypes and operation centers [8,9,10,11,12,13]. Among them, PFIC type I patients suffered a relatively lower 10-year survival rate than other PFIC types, as low as 58% in a study [9]. This is often related with the refractory diarrhea and the high frequency of steatohepatitis in PFIC type I patients after LT.

The present study aims to evaluate the long-term outcome of PFIC patients after LT, including the survival rate, development, and post-LT complications. By collecting the medical records and follow-up files, we retrospectively analyzed how LT improved the liver function, physical and intellectual development, and quality of life in PFIC patients. Meanwhile, complications after LT, especially diarrhea and steatosis in PFIC type I patients remind us to maintain close monitoring during long-term follow-ups. 

## 2. Patients and Methods

From October 2007 to May 2019, 41 PFIC patients received LT in Ren Ji Hospital affiliated to Shanghai Jiao Tong University School of Medicine. The diagnosis was based on clinical manifestations, laboratory results, and genetic testing. Patients’ characteristics were recorded at the time of LT, and comprehensive surgical data, complications, and follow-up data after LT were also collected and analyzed, with the last clinical visit in May 2022. The pre-LT evaluation, surgical procedures, postoperative immunosuppression strategy, and donor evaluation have previously been reported [14]. The follow-up time ranged from 3 years to 15 years. To monitor the development of steatohepatitis after LT, blood tests, ultrasound, and computed tomography (CT) scan were used as screening methods, and for patients who were suspected with fatty changes in the graft; liver biopsy was recommended. An evaluation of physical and intellectual development after LT was conducted using the Ages and Stages Questionnaires (ASQ) that was finished by the patient’s parents. Height and weight Z scores were also applied to assess the physical development after LT. 

Statistical analyses were conducted with SPSS software, version 23.0 (IBM, Armonk, NY, USA), and Prism software, version 8.4.0 (455) (Irvine, CA, USA). Survival analyses were evaluated using the Kaplan–Meier method starting from the date of LT to the last clinical visit or the date of death. Continuous variables, such as age, were shown as median with interquartile range (IQR), while categorial variables, such as blood type, were shown as ratio or percentage. A *p*-value less than 0.05 was considered statistically significant. 

## 3. Results

Among 41 PFIC patients who received LT in our center, 4 patients were diagnosed as PFIC type I with genetic mutation of *ATP8B1*, 25 as PFIC type II with mutated *ABCB11,* and 12 as PFIC type III with *ABCB4* mutation (Table 1). All patients received LT due to the presence of end-stage liver disease, with eight patients being accompanied by severe pruritus. The median Child–Pugh score at LT was B10 (B7-C13). Before transplantation, pulmonary artery stenosis presented in 12.2% of patients, while cholangitis happened in 14.6%. Atrial septal defect (4.9%), ventricular septal defect (2.4%), and gastrointestinal bleeding (9.8%) were also observed. Surgical intervention was performed in nine patients before LT, including six patients who received partial internal biliary diversion (PIBD) and three who received the Kasai procedure (Appendix A). UDCA (ursodeoxycholic acid) was the most used drug therapy in PFIC patients before LT. Relational analysis indicated that the serum bile acid level before transplantation has no effect on the age of LT (Appendix A). 

At the time of LT, the median age was 2.93 (0.75–5.98) years, with 75.6% of patients receiving living donor liver transplantation (LDLT) in which donors were their parents. The type of graft was left lateral lobe in 48.8%, left liver in 24.4%, and whole liver in 19.5%. Median intraoperative blood loss was 200 mL, and the median postoperative hospital stay was 20 days (Table 1). Following transplantation, the liver function and bile acid metabolism returned to a normal level around two weeks after LT, with relieved jaundice and improved coagulopathy condition. Median serum bile acid decreased from 200.9 μmol/L before LT to 10.3 μmol/L after LT, and pruritus was cured in all patients. The 5- and 10-year patient survival rates after LT were 92.7% and 92.7%, respectively. No survival difference was found among the three subtypes of PFIC patients. The 5- and 10-year graft survival rates were 87.8% and 87.8% (Figure 1). Two PFIC type II patients received re-transplantation at 5 months and 2.8 years after their first transplantation due to hepatic vein complication and hepatic artery complication, respectively. Mortality occurred in three patients (one PFIC type II patient and two PFIC type III patients) within one year after LT, which resulted from severe pulmonary infections.

Catch-up growth and improvement of physical development were reported after LT. The mean height Z score improved from −2.53 to −0.55 with a median follow-up time of 5.55 years, and the mean weight Z score increased from −1.54 to −0.27 (Figure 2). Meanwhile, psychomotor skills improvement was also found in all recipients during follow-up, with mild retardation in six patients. Among them, one patient suffered from causal uncoordinated movements, while the other five patients encountered mild muscle growth retardation. No abnormalities in gross motor skills and fine motor skills were reported.

Academic performance was analyzed for school-aged patients. A total of 29/30 (96.7%) of school-aged patients had attended school after LT. Average academic performance was observed in 72.4% of school-aged patients. One patient successfully passed the National College Entrance Exam and started her college study at 18 years old, which was 4 years after LT. One patient dropped out from elementary school due to repeated fatigue symptoms and emotional problems. Retardation in learning ability was also reported, which might be related to the potential encephalopathy-related brain damage before LT. 

In our study, all four PFIC type I patients reported diarrhea after LT, happening between 3 to 14 months. Probiotics, cholestyramine, and supportive treatments were used with a satisfactory effect. Normal stool output and nutritional status were reported during follow-up. Among them, steatosis in the graft was observed in one patient 15 months after LT, who received the left liver from his parent at the age of 6.6 years old. Since no diarrhea or malnutrition was reported afterward, this patient received no additional treatment but cholestyramine. Attention should be paid to the fact that steatosis was relieved 36 months after LT, reappeared at 50 months, and was relieved again at 77 months post-LT (Figure 3A). However, no relationship between the changes and the fluctuations in serum bile acid (BA), γ-glutamyl transferase (GGT), and transaminase (ALT) was found during the above process (Figure 3B). Meanwhile, no hypoproteinemia, malnutritional enteropathy, or pancreatitis was detected during the follow-up, and improved physical development was also observed, as the height Z score increased from −1.51 before transplantation to 0.38 nine years after LT. The other three PFIC type I patients never developed steatosis during follow-up, with normal liver function and physical development.

## 4. Discussion

Progressive familial intrahepatic cholestasis (PFIC) describes a group of diseases that represents defects in the bile acid synthesis and handling [5,15]. Once progressed to ESLD, LT is the only effective treatment to cure clinical symptoms and improve the survival rate. In this study, we analyzed the long-term survival rate and development after LT. Firstly, a satisfactory long-term survival rate after LT was observed, with 5- and 10-year patient survival rates of 92.7% and 92.7%, respectively. No noticeable discrepancy was found among the three subtypes of PFIC. Secondly, catch-up growth and improved physical and intellectual development were recorded after LT, with improved quality of life. Thirdly, diarrhea after LT was found in all four PFIC type I patients but was well controlled after bile absorptive resin treatment. However, one patient still encountered a “remission–relapse circle” of steatosis in allograft during follow-up. No relationship is found between the status of steatosis and fluctuations of liver enzymes.

PFIC was previously divided into three types based on the mutated genes of *ATP8B1*, *ABCB11*, and *ABCB4* [7]. However, with the development of diagnostic methods, such as next-generation sequencing and whole-exome sequencing, new mutated genes have been detected in recent years, such as *TJP2*, *NR1H4*, and *MYO5B* that are responsible for PFIC type IV, V, and VI, respectively [2]. Even drug therapy and surgical procedures such as biliary diversion could partially delay the progression of PFIC, most PFIC patients need to receive LT treatment due to ESLD, refractory pruritus, or hepatocellular carcinoma [16]. The reported overall one-year survival rates of PFIC patients after LT range from 64% to 100%, varying among subtypes and operation centers [9,17,18]. However, reports of the long-term survival rates with large study populations are relatively rare. In our study, we focused on the long-term survival rates. We found that the overall 10-year survival rate after LT is 92.7%, which is comparable to biliary atresia or other cholestatic disease recipients. What is more, no difference of survival rate was found between subtypes. Consistent with previous studies, no recurrence of cholestasis or development of liver tumor was reported during the long-term follow-up in PFIC type II and type III patients, while allograft steatosis was observed in one PFIC type I patient [16]. Therefore, LT remains one of the most efficient treatments for PFIC patients with ESLD.

Whether liver transplantation is suitable for PFIC type I patients is controversial. Extrahepatic manifestations in PFIC type I patients, such as watery diarrhea and short stature may persist after LT, which may require additional surgical intervention or medications [6]. Besides, steatohepatitis after LT could lead to cirrhosis and require re-transplantation. Our study and several previous studies show that LT improves symptoms and survival rate irrespective of PFIC subtypes [6,8,10]. Due to the dysfunction of SLC10A2 in the apical membrane of PFIC type I patients, bile acid reabsorption in the intestine is disrupted, especially when the explanted graft starts to produce and secrete a normal volume of bile acids after LT [19]. As reported, 30–70% of PFIC type I patients encountered allograft steatosis after LT, most of which would develop into steatohepatitis and cirrhosis, and require re-transplantation [20]. Bile absorptive resin is shown to be capable of reducing diarrhea and steatosis after LT, but the effect is considered genotype dependent [7]. LT combined with biliary diversion may be of help to revert steatohepatitis after LT, with liver–intestine co-transplantation is also proposed [19,21,22,23,24]. In our study, one PFIC type I patient who received LDLT reported diarrhea 6 months after LT and began to intake cholestyramine, which resulted in normal fecal output. However, steatosis in allograft was found 15 months after LT. Although no modification of the treatment strategy was made, the steatosis was relieved around 36 months after LT, then reappeared and was alleviated again at 60 months and 77 months after LT. The “remission–relapse circle” of allograft steatosis after LT in PFIC patients was barely discussed before, and no connection between steatosis and liver enzyme level was found in our case. Since no diarrhea or malnutrition was found after cholestyramine treatment in this patient, other factors that may cause allograft steatosis should be investigated. Meanwhile, for patients whose diarrhea is well controlled by medical treatment or surgical intervention, routine abdominal CT examination or biopsy should be considered, as allograft steatosis may relapse without causing liver dysfunction. The other three PFIC type I patients, among which two received orthotopic liver transplantation (OLT) and one received LDLT, also encountered diarrhea after LT but were well-treated by probiotics and cholestyramine. No allograft steatosis was found in them during the follow-up. 

Improvement of development and life quality is also an important research field after pediatric LT. In our study, catch-up growth and improvement in psychomotor ability and intelligence development were observed in all PFIC patients after LT. One patient started her college study four years after LT with impressive academic performance. However, retardation in physical development and academic performance was still reported. The dominant manifestations included amnesia, muscular atrophy, retarded mathematical ability, delayed language skills development, and ADHD-like manifestations. It was reported that clinical symptoms before LT had a significant influence on the post-LT development in PFIC patients [25]. Pruritus before LT causes irritability, loss of sleep, and poor attention, which could result in impaired school performance and brain damage [1]. Meanwhile, impaired vitamin D and calcium absorption resulting from decreased bile secretion also led to growth retardation and metabolic bone diseases [26]. LT improves physical development and life quality by resolving cholestasis and pruritus, facilitating vitamin D absorption, increasing insulin-like growth factors production and relieving portal hypertension [27]. However, extrahepatic manifestations and complications after LT also affect the physical growth [22]. Retardation in academic performance was thought to be related to absence from school, lack of exercise, and latent brain damage before LT. A study from ChiLDReN (Childhood Liver Disease Research Network) stated that PFIC patients had no risks of lower Full Scale Intelligence Quotient (FSIQ) when compared with population norms. However, lower albumin was associated with lower FSIQ in PFIC patients with native liver, suggesting that the seemingly delayed intelligence development may be influenced by malnutrition and liver dysfunctions, regardless of original disease [28]. Therefore, nutritional status plays a key role in both physical and intellectual development.

We are aware of the limitations of our retrospective study. One limitation is the relatively small sample size due to the rarity of PFIC patents who need LT treatment, especially type I patients. Meanwhile, the lack of genotypes in some patients limits our further analysis of its relationships with long-term survival rates, complications, and development. Besides, some patients received the Kasai procedure before being referred to our center without genetic mutation tests, which may result in the LT operation occurring earlier than expected. However, the encouraging long-term survival rates of PFIC patients after LT reminds us that for PFIC in ESLD or with refractory pruritis, LT should be conducted at an optimal time. The personalized treatment strategy should be made based on comprehensive and precise assessment at early stage.

All in all, our study witnessed a good outcome of PFIC patients who underwent LT. A total of 41 PFIC patients received LT with a 5- and 10-year survival rate of 92.7% and 92.7%, respectively, and no difference was found between subtypes. Catch-up growth and improved physical and intellectual development were observed during follow-up, with significantly improved life quality. One PFIC type I patient exhibited a “remission–relapse circle” of allograft steatosis after LT with the treatment of cholestyramine, whereas the liver function maintained normal. Taken together, LT is an efficient treatment for PFIC patients with a satisfactory long-term survival rate and life quality. 

## 5. Conclusions

In summary, our study indicates that LT is an effective treatment for end-stage PFIC patients with a satisfactory long-term survival rate. Physical and intellectual development after LT guarantees an improved life quality and academic performance in those patients, although mild retardation is observed in some patients. Meanwhile, allograft steatosis should be closely monitored in PFIC type I patients after LT, even if refractory diarrhea is well-treated.

## Figures and Tables

**Figure 1 jcm-11-04684-f001:**
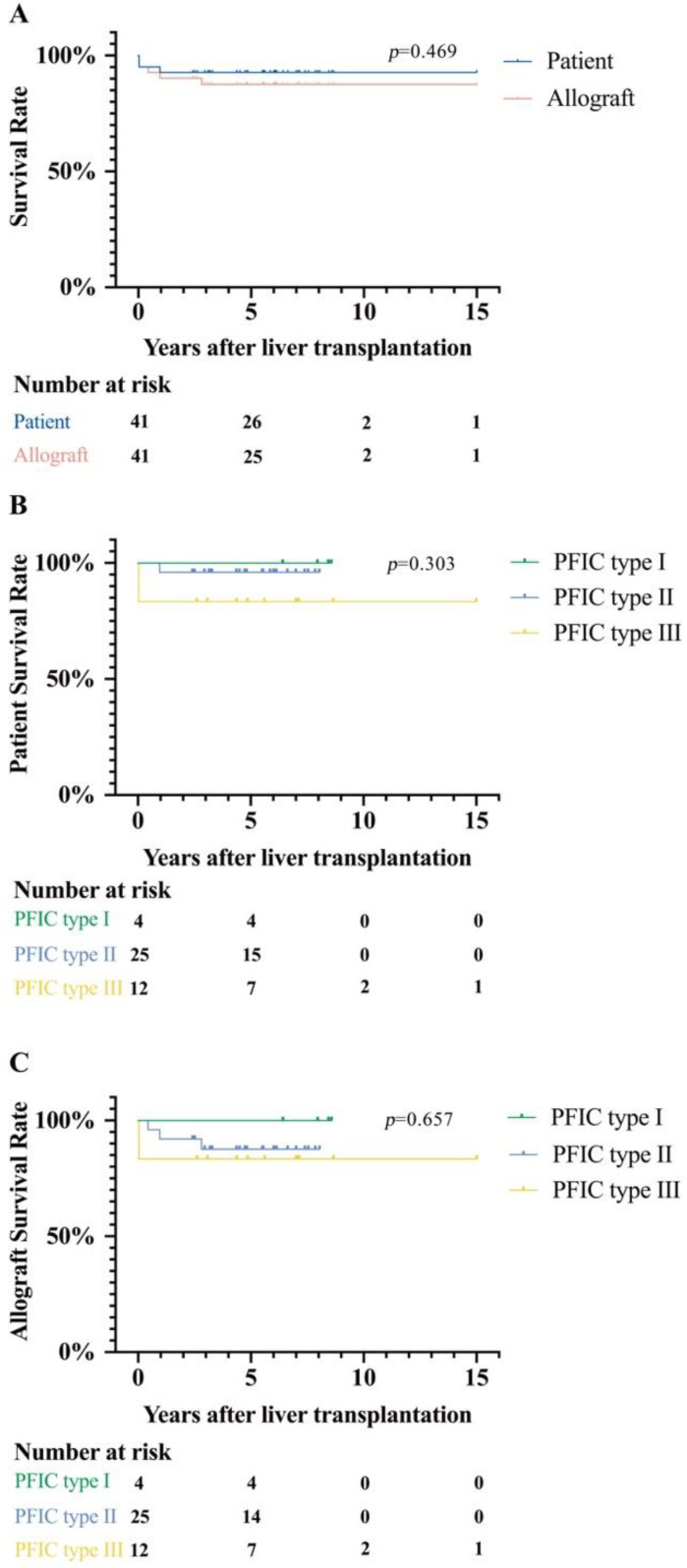
Patient and allograft survival rates after LT. (**A**) Overall PFIC patient survival rates were 92.7% and 92.7% at 5 and 10 years after LT, respectively. The longest follow-up time is 15 years after LT. Allograft survival rates of PFIC patients were 87.8% and 87.8% at 5 and 10 years, respectively. (**B**) Patient survival rates of three subtypes of PFIC patients. PFIC type I patient survival rate (4 patients in total) was 100.0% at 5 years after LT. PFIC type II patient survival rate (25 patients in total) was 96.0% at 5 years after LT. PFIC type III patient survival rate (12 patients in total) was 83.3% at 5 years after LT. (**C**) Allograft survival rates of three subtypes of PFIC patients.

**Figure 2 jcm-11-04684-f002:**
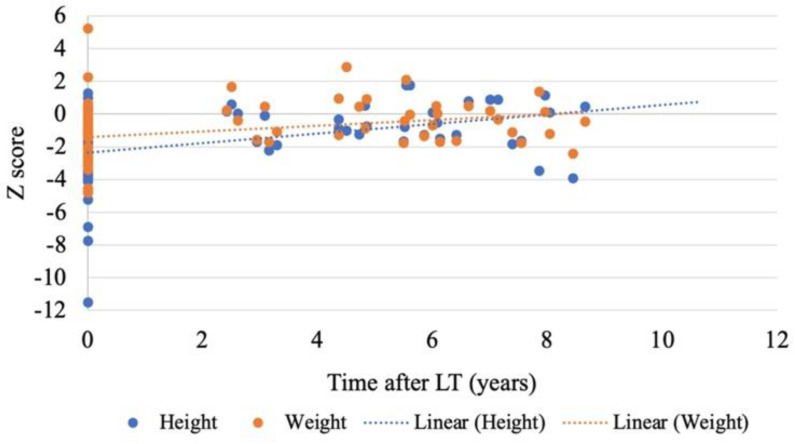
The changes of height and weight Z score after LT. Catch-up growth after LT was witnessed in PFIC patients. The mean height Z score improved from −2.53 to −0.55 with a median follow-up time of 5.55 years, and the mean weight Z score increased from −1.54 to −0.27.

**Figure 3 jcm-11-04684-f003:**
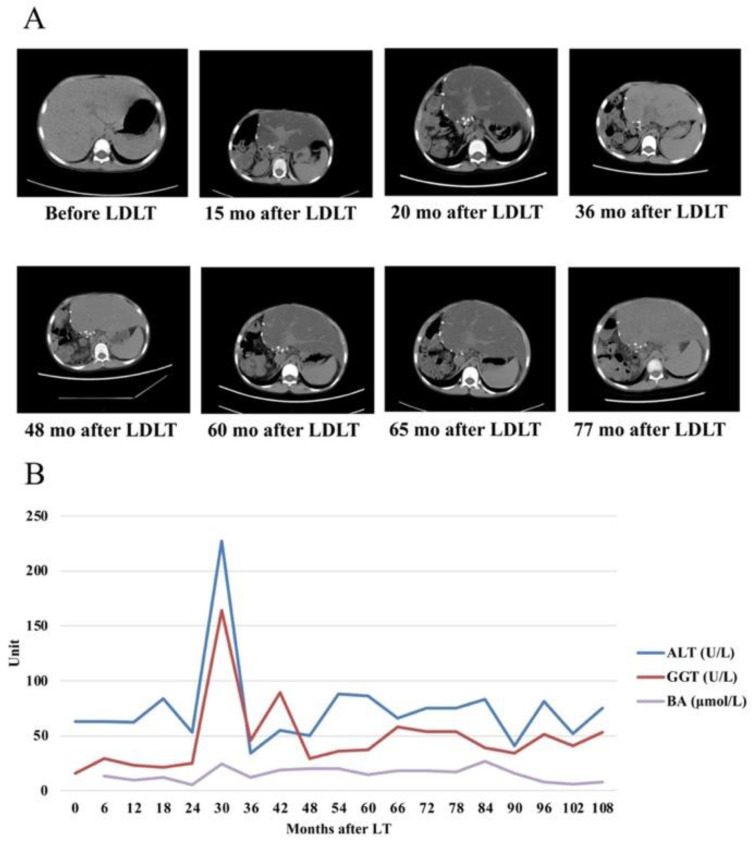
Allograft steatosis and liver function in one PFIC type I patient after LT. (**A**) Abdominal computed tomography (CT) examination before and after LT in one PFIC type I patient indicated the “remission–relapse circle” of steatosis in the allograft. Steatosis was firstly found 15 months after LT and was relieved at 36 months but reappeared and alleviated again at 60 months and 77 months, respectively. (**B**) Liver enzymes fluctuation has no relation with allograft steatosis. LDLT: living donor liver transplantation; ALT: alanine aminotransferase; BA: bile acid; GGT: γ-glutamyl transferase; mo: month.

**Table 1 jcm-11-04684-t001:** Basic clinical characteristics of the PFIC patients.

Characteristics	PFIC Patients (*n* = 41)
Basic information	
Gender (M/F)	30/11
Median age at LT (year)	2.93 (0.75–5.98)
Blood type (A/B/O/AB)	17/8/16/0
Subtypes of PFIC	
PFIC type I	9.8% (4/41)
PFIC type II	61.0% (25/41)
PFIC type III	29.3% (12/41)
Surgical information	
Type of allograft	
Left lateral lobe	48.8% (20/41)
Left liver	24.4% (10/41)
Whole liver	19.5% (8/41)
Right liver	2.4% (1/41)
Right posterior lobe	2.4% (1/41)
Expanded left lateral lobe	2.4% (1/41)
Surgical procedure (LDLT/DDLT)	31/10
Median intraoperative blood loss (mL)	200 (100–300)
Median hospital stays post-LT (day)	20 (15.25–32.75)
Median GRWR	2.47% (1.86−3.10%)
Immunosuppressive strategy	
Tacrolimus	87.8% (36/41)
Cyclosporine	12.2% (5/41)

DDLT, deceased donor liver transplantation; GRWR, graft to recipient weight ratio; LDLT, living donor liver transplantation; LT, liver transplantation.

## Data Availability

Not applicable.

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
