# Peer review of "Long-Term Results of Pediatric Liver Transplantation for Progressive Familial Intrahepatic Cholestasis"

_jcm, 2022, doi:10.3390/jcm11164684_

Round 1

Reviewer 1 Report

Introduction

In this section, the authors present the study results and conclusions, which is incorrect. They should present only the study aim.

Patients and methods

Patients characteristics should be reported in the "Results" section, not in the "Patients and methods" section.

The methods for monitoring the possible development of steatosis after LT should be reported.

Statistical methods should be better explained (e.g., the difference in presenting continuous and categorical variables).

Results

The sentence "[...] psychomotor skills improvement was also reported by the children's patients, with mild retardation in several patients" needs revision. Who reported psychomotor skills improvement? Moreover, the number of patients with mild retardation should be detailed.

The sentence "Diarrhea and steatosis are the most common complications in PFIC type 1 patients after LT" is inappropriate for the "Results" section. The adequate setting for considerations on background is represented by the "Introduction" or "Discussion" section.

Figure 1 should include the number of patients at risk. Moreover, P values should be included in figure 1 B and C.

Discussion and conclusions

The authors state that no tumor recurrence was observed. However, when they describe the study cohort, they do not disclose cases of liver tumors.

The limitations of this study should be reported. Although justified by the rarity of the disease, the small sample size does not allow to draw any conclusions, especially regarding the outcome of PFIC type 1 patients (only 4 cases).

Reviewer 2 Report

The authors present a single-centre long-term analysis over 12 years of outcome data on liver transplantation for end-stage liver disease in 41 patients with progressive familial intrahepatic cholestasis (PFIC). The emphasis is upon patient and graft survival, and anthropometric and gross cognitive outcome data. The authors present an excellent 5- and 10-year-survival rate for both patients and graft. Additionally, catch-up growth, improved psychomotor ability and age-appropriated study ability are recorded.

The term PFIC comprises an ever-expanding list of diseases with complex pathophysiology and heterogeneous clinical presentation. Throughout the manuscript, the knowing reader gets the impression, that the authors only superficially touch the current understanding of PFIC, which results in erroneous statements and citations (examples will be given later).

At its current state the manuscript is a mere enumeration without deeper use to the understanding of the PFIC spectrum. PFIC is rare, therefore every publication should aim at expanding knowledge which will aid to offer the best treatment options for patients affected. Particularly, follow-up data to liver transplantation is highly relevant.

Still, I consider this manuscript to be relevant but major additions and corrections should be met prior to publication.

Major comments:

The authors show survival for each PFIC1-3. Recent work of the NAPPED consortium has nicely shown, that the genotype strongly impacts on the native liver survival of PFIC1- and 2-patients. Additionally, not only the genotype but also serum bile acids have predictive value on the native liver survival.

An extensive table on genotype and individual patient characteristics would strongly improve the manuscript. I consider this worthy of a table within the main manuscript but if not otherwise possible, at least as supplementary table. Here, the authors could include extra-intestinal manifestations, parameters of end-stage liver disease (including serum bile acids), surgical and/or pharmaceutical therapy preceding liver transplantation (LT), parameters of LT, post LT serum bile acids, histopathology on liver biopsies in case of steatosis for PFIC1 patients, frequency/consistency of diarrhoea, treatment, … should be included. Also, encephalopathy due to end-stage liver disease should be reported for each individual patient, as this is used for reasoning in line 132-133.

In one of four PFIC1 patients, steatosis after LT is reported. Additionally, elevated and undulating liver enzyme tests (mistaken and wrongly worded as “liver function” in line 155 and line 171) are shown. This would suggest graft inflammation, taken together as steatohepatitis. Was a biopsy performed to further distinguish steatohepatitis from rejection? If so, please add the histopathological details, as mentioned in the paragraph above.

This would help categorization of PFIC1 genotypes with post-LT complications. The same holds true for PFIC2, as autoimmunization against the novel BSEP protein after LT is reported. Here, matching genotypes with characteristics of donors (heterozygosity,…) would be valuable.

Accordingly, the authors could calculate survival rates, as shown in Figure 1, for genotypic subgroups of PFIC1-3, if this is possible due to group size and state, whether this led to the same of different results. Furthermore, I advise to include the “number at risk” at the bottom of the individual Kaplan-Meier curves. This would help the reader to interpret the data.

The information on pulmonary artery stenosis, cholangitis, atrial septal defect, ventricular septal defect and gastrointestinal bleeding can be also included into the suggested table.

The statement in line 187 “Therefore, LT remains the most efficient treatment for PFIC patients.” must be toned down.

Biliary diversion after LT was shown to effectively revert steatohepatitis by several groups. Here, this is referred to as “LT combined with biliary diversion may be of help to slow down to relapse of steatosis…”. Steatosis seems to be used interchangeable with steatohepatitis. I recommend rephrasing of this sentence.

Minor comments:

-) Patients and Methods: If an ethical statement would be required, it is missing.

-) When a median is presented, it is advised to state the interquartile range (IQR)

-) The authors should also state total follow-up years

-) line 39: “Based on genetic mutations…”. Within the literature PFIC6 instead of “PFIC associated with MYO5B defects” seems to be the commonly accepted term.

-) line 41: The sentence “All subtypes of PFIC are caused by defects in bile secretion from hepatocyte to bile canaliculi” is both orthographically and substantially incorrect. PFIC4 (TJP2-deficiency) seems not have an underlying secretory defects. A similar statement at the beginning of the Discussion (line 160) needs correction as well.

-) Abstract: abbreviations are used without prior introduction (PFIC, LT,…)

-) line 20: “2.93-year-old” should read “2.93 years”

-) line 47: I would not know which diet is a proper treatment for PFIC. Maybe the authors could elaborate on this a little more.

-) line 48: exchange “like” for “such as”

-) line 54: “One possible reason is the high frequency of extra-hepatic complications after LT.” This statement may be correct for intractable diarrhoea after liver transplantation but this seems the be most likely the only extra-hepatic reason. Another possible reason is the intra-hepatic complication, being a steatohepatitis, which necessitates re-transplantation if kept unchecked.

-) line 136: The authors should elaborate on the antibiotics used to treat diarrhoea post-LT.

-) line 141: “What’s interesting,…” is no appropriate phrasing for a scientific manuscript

-) line 197: The statement “In this case, osmotic diarrhea…” is not stated in and supported by the citation given. The phenomenon underlying steatohepatitis after LT for PFIC1 is far from understood. The sentence should be removed.

-) line 241: The conclusion drawn from reference #29 should be carefully revisited, as this work included patient with native livers and albumin might not only be expression of nutrition but of liver function as well.

Round 2

Reviewer 2 Report

The authors have submitted a revised and improved version of the manuscript "Long-term Results of Pediatric Liver Transplantation for Progressive Familial Intrehepatic Cholestasis". Previous comments, remarks and suggestions were adressed.

The now added supplementary table 1 provides details to each individual patient but also highlights a flaw of the study to be still present - the missing genotypes of many patients. The authors discuss this matter. Still I would recommend this manuscript for publication, when some minor corrections are addressed:

-) "All subtypes of PFIC are caused by defects in bile acid synthesis and handling" This sentence in the introduction and discussion is not correct. Please rephrase to "intracellular bile acid transport, secretion and homeostasis of bile canalicular function".

-) The authors have added the numbers at risk to the Kaplan-Meier curves in Figure 1. Please also add the p-value, despite it probably being insignificant.

-) The authors have changes "steatosis" to "fatty changes" when reporting one PFIC1-patient. Maybe my comment was misleading. Please change it back to steatosis, as I assume it was stated in the CT-findings and also add the source of the finding (CT or biopsy).

-) Please also add a column in supplementary table, were steatosis is indicated for the respective PFIC1-patient.

-) Supplementary table 1 yields follow-up liver enzyme test that show elevated AST/ALT in several patient. Could the authors comment on that, whether this is graft inflammation, cholangitis, rejection, or not known... in an additional comment column perhaps?

-) line 238: Please rephrase to: "... one of the most efficient treatment for PFIC patients with ESLD."

-) A mere idea, which the authors may choose to follow is the integration and discussion of the correlation analysis of Figure R1 into the main manuscript.
